# Quantifying the roles of vomiting, diarrhea, and residents vs. staff in norovirus transmission in U.S. nursing home outbreaks

**Carly Adams**[1]*, **David Young**[2], **Paul A. Gastañaduy**[3], **Prabasaj Paul**[4], **Zach Marsh**[3,5], **Aron J. Hall**[1,3], **Benjamin A. Lopman**[1]

**1** Department of Epidemiology, Rollins School of Public Health, Emory University, Atlanta, Georgia, United States of America, **2** South Carolina Department of Health and Environmental Control, Columbia, South Carolina, United States of America, **3** Division of Viral Diseases, National Center for Immunization and Respiratory Diseases, Centers for Disease Control and Prevention, Atlanta, Georgia, United States of America, **4** Division of Healthcare Quality Promotion, Centers for Disease Control and Prevention, Atlanta, Georgia, United States of America, **5** Oak Ridge Institute for Science and Education, Oak Ridge, Tennessee, United States of America

* carly.adams@emory.edu

**Data Availability Statement:** All relevant data are within the manuscript and its Supporting Information files.

## Abstract

The role of individual case characteristics, such as symptoms or demographics, in norovirus transmissibility is poorly understood. Six nursing home norovirus outbreaks occurring in South Carolina, U.S. from 2014 to 2016 were examined. We aimed to quantify the contribution of symptoms and other case characteristics in norovirus transmission using the reproduction number ($R_{Ei}$) as an estimate of individual case infectivity and to examine how transmission changes over the course of an outbreak. Individual estimates of $R_{Ei}$ were calculated using a maximum likelihood procedure to infer the average number of secondary cases generated by each case. The associations between case characteristics and $R_{Ei}$ were estimated using a weighted multivariate mixed linear model. Outbreaks began with one to three index case(s) with large estimated $R_{Ei}$'s (range: 1.48 to 8.70) relative to subsequent cases. Of the 209 cases, 155 (75%) vomited, 164 (79%) had diarrhea, and 158 (76%) were nursing home residents (vs. staff). Cases who vomited infected 2.12 (95% CI: 1.68, 2.68) times the number of individuals as non-vomiters, cases with diarrhea infected 1.39 (95% CI: 1.03, 1.87) times the number of individuals as cases without diarrhea, and resident-cases infected 1.53 (95% CI: 1.15, 2.02) times the number of individuals as staff-cases. Index cases tended to be residents (vs. staff) who vomited and infected considerably more secondary cases compared to non-index cases. Results suggest that individuals, particularly residents, who vomit are more infectious and tend to drive norovirus transmission in U.S. nursing home norovirus outbreaks. While diarrhea also plays a role in norovirus transmission, it is to a lesser degree than vomiting in these settings. Results lend support for prevention and control measures that focus on cases who vomit, particularly if those cases are residents.

**Funding:** This work was supported by NIH/NIGMS (R01 GM124280) and by AHRQ (R01 HS025987). The funders had no role in study design, data collection and analysis, decision to publish, or preparation of the manuscript. The corresponding author had full access to all data in the study and had final responsibility for the decision to submit for publication.

**Competing interests:** I have read the journal's policy and the authors of this manuscript have the following competing interests: BAL reports personal fees from Takeda Pharmaceutical, personal fees from CDC Foundation, and personal fees from Hall Booth Smith, P.C., outside the submitted work.

## Author summary

The majority of all norovirus outbreaks reported to the CDC occur in long-term care facilities (LTCFs), including nursing homes, where older residents are at risk for more severe or prolonged infection. Because there is currently no publicly available norovirus vaccine, sound control measures are key to controlling norovirus outbreaks, but there is little evidence that standard control measures are effective in reducing the size and/or duration of LTCF norovirus outbreaks. Hence, studies leading to a better understanding of disease spread and prevention of additional cases, and thus more effective control measures, are needed. To this end, we aimed to quantify factors associated with norovirus transmission and to examine how transmission changes over the course of an outbreak. We show that vomiting and, to a lesser extent, diarrhea are critical in initiating and sustaining norovirus transmission in U.S. nursing home norovirus outbreaks. We also show that nursing home residents, rather than staff, are the primary drivers of transmission. Results suggest that control measures focusing on cases who vomit, particularly if those cases are residents, would be most effective at curtailing norovirus transmission in these settings.

## Introduction

There are 49.2 million individuals over 65 in the U.S. population (15.2%) and this population is growing [1]. With nearly half of this age group spending some part of their lives in nursing homes [2], the number of older adults using paid long-term care services is expected to grow substantially over the coming decade [3]. In the U.S. and other high-income countries, gastroenteritis outbreaks are common in nursing homes, skilled nursing facilities and assisted living facilities, which are collectively known as long-term care facilities (LTCFs) [4–7]. Despite the perception that norovirus is a foodborne disease or the 'cruise ship virus', the majority of all norovirus outbreaks reported to the CDC occur in LTCFs [6]. While norovirus gastroenteritis is generally mild and self-limiting, older nursing home residents are vulnerable to infection leading to hospitalization and death [8], with the vast majority of norovirus-associated deaths in the U.S. occurring among persons aged 65 years and older [9].

Norovirus is highly transmissible in nursing homes [10–12], but there is no vaccine or specific antiviral therapy available to prevent or treat norovirus infection. As a result, rapid implementation of standard control measures is the mainstay for curtailing transmission [13]. Identifying factors associated with norovirus transmission is critical to better understanding disease spread and preventing additional cases. Individual-level risk factors for susceptibility to norovirus infection or severe disease in nursing home outbreaks have been identified, including resident mobility, dependency on staff assistance [14], immunodeficiency [15], and statin use [16]. But because transmission of norovirus from one person to another cannot be directly observed (unlike symptoms and/or positive test results that follow transmission), it remains poorly understood and the evidence base for the value of specific prevention and control measures is lacking [10].

Statistical algorithms can be used to infer outbreak transmission trees (i.e., who infected whom) from case onset dates and independent estimates of the serial interval (i.e., the time between symptom onset in primary cases and the secondary cases they generate) between generations of case pairs [17]. Individual reproduction numbers ($R_i$), or the number of secondary cases an individual generates, can then be calculated for all cases. We quantified the contribution of specific symptoms and residents vs. staff in norovirus transmission by examining the associations between these variables and individual case infectivity, which was characterized

by $R_i$. Additionally, we examined how transmission changes over the course of an outbreak. Our overall aim was to inform implementation of effective norovirus prevention and control measures to reduce the size and duration of norovirus outbreaks in nursing homes. We achieved this aim by characterizing norovirus transmission in these settings.

## Methods

### Ethics statement

As this was an analysis of anonymized data that had already been collected through routine public health response, the Emory University Institutional Review Board (IRB) determined that this study was exempt from IRB review.

### Outbreak data

De-identified data from six separate and unique nursing home outbreaks from two consecutive norovirus seasons (December–April, 2014–2015 and 2015–2016) were provided by the South Carolina Department of Health and Environmental Control (SCDHEC) (S1 File). All outbreaks were confirmed, meaning they had at least two laboratory confirmed norovirus cases. Outbreak data were in the form of line lists and included individual-level information on symptom onset dates, reported symptoms (vomiting, diarrhea, and fever), age in years, sex, illness duration, hospitalization, emergency department visit, and whether the case was a resident or staff. Probable cases were defined as residents or staff who had at least one episode of vomiting and/or three or more loose stools within a 24-hour period. Confirmed cases were probable cases with a laboratory confirmed norovirus infection.

### Estimation of reproduction numbers

Transmissibility of a pathogen can be quantified by its basic reproduction number, $R_0$, defined as the average number of secondary cases generated by a single infectious individual in a population that is entirely susceptible, or its effective reproduction number, $R_E$, defined as the average number of secondary cases generated by a single infectious individual in a population that has some level of immunity. $R_0$ or $R_E$ of 1 signifies the extinction threshold, below which each infectious individual, on average, infects less than one other individual and the outbreak cannot be maintained. $R_E$ can be converted to $R_0$ by dividing $R_E$ by the proportion susceptible in the population. Estimates for the $R_0$ and $R_E$ of norovirus vary widely (1.1 to 7.2 and 0.85 to 14.05, respectively), and depend on differences in settings [18].

The primary outcome of interest in this study was individual case infectiousness, which we measured by estimating the reproduction number, $R_{Ei}$, for each case. Here, $R_{Ei}$ is defined as the number of secondary cases generated by an individual case $i$. We estimated $R_{Ei}$ using a maximum likelihood procedure to infer the number of secondary cases generated by each case (S1 RMarkdown File) [17]. This method, originally described by Wallinga and Teunis, requires only onset dates of all cases in the outbreak and knowledge of the probability distribution of the serial interval for the specific infectious disease [17]. We used a serial interval for norovirus derived from several large norovirus outbreaks in child daycare centers in Sweden with a gamma probability distribution, mean of 3.6 days, and standard deviation of 2.0 days [19]. We performed sensitivity analyses with mean serial intervals varying between 1.5 and 4.0 days in half day increments (S2 RMarkdown File). Details of the estimation procedure are available elsewhere [17,19,20]. Briefly, this method uses the difference in symptom onsets dates between cases and the probability distribution of the serial interval to calculate the relative likelihood that cases with earlier symptom onset dates infected cases with later symptom onset dates. The

relative likelihoods are then summed to estimate the $R_{Ei}$ for each case. Individual cases were assigned a $R_{Ei}$ and corresponding 95% confidence interval based on their symptom onset date, and those with the same onset date within an outbreak were assigned the same $R_{Ei}$ and confidence interval.

In preliminary analysis, we observed much higher $R_{Ei}$ for index cases compared to those on subsequent days. To investigate whether this could indicate heightened infectiousness of index cases or just the natural decline of the susceptible population, we also calculated $R_{0i}$ by dividing $R_{Ei}$ by the proportion of the population susceptible on day $i$ ($p_i$) [21]. To calculate the proportion susceptible, we made the extreme assumptions that all cases were susceptible at the start of

the outbreak and that the final cumulative attack rate was 100%, such that $p_i = 1 - \frac{\sum_1^i C_i}{C_1}$

where $C_1$ is the total number susceptible on day 1, or the total number of cases in the outbreak minus the number of index cases, and $\sum_1^i C_i$ is cumulative incidence to day $i$ (excluding index cases). Using this approach, we compared estimates of $R_{0i}$ of index cases on day 1 to $R_{0i}$ estimated from cases with onset on days 2 to 4 of the outbreak (excluding days with no reported cases).

## Analyses of risk factors for transmission

We used a linear mixed model to estimate the association between each case characteristic and $R_{Ei}$, while accounting for correlation between $R_{Ei}$'s within each outbreak (S3 RMarkdown File). The outcome variable was the natural log of $R_{Ei}$.

The following information was available for cases: symptom onset date, resident/staff status, age in years, sex, illness duration, hospitalization, emergency department visit, and presence of diarrhea, vomiting, and fever. Because information on fever, age, sex, emergency department visit and hospitalization were missing for large percentages of cases (20%, 23%, 26%, 40% and 55%, respectively), we were unable to consider these variables as potential exposure, confounder, or effect modifying variables in the regression model. However, a sensitivity analysis was performed using cases with available information on age and/or sex to examine whether these could be potential confounding variables. Information on resident vs. staff, diarrhea (yes or no), and vomiting (yes or no) were rarely missing (1%, 1%, and 0%, respectively) and were considered explanatory variables in our model. To account for clustering induced by correlation of $R_{Ei}$'s within the six outbreaks, outbreak number was included in the model as a random intercept. The full model, with log $R_{Ei}$ as the outcome, included the following explanatory variables: diarrhea, vomiting, resident. Furthermore, because each $R_{Ei}$ estimate had its own uncertainty, we used a meta-analysis approach and incorporated $R_{Ei}$ uncertainties by using inverse variance weighting. Weights were equal to the inverse of the sum of the three variance components: 1. variance from estimation of $R_{Ei}$ (unique to each onset date), 2. within-outbreak variance (unique to each outbreak), and 3. between-outbreak variance (equal for all estimates). The model was assessed for collinearity and no issues were found. We considered including 'time' in the model and adjusting for it as a potential confounder, as $R_{Ei}$ inevitably declines over time. However, we determined that time cannot be a confounder, since it cannot affect diarrhea, vomiting, or resident vs. staff, our explanatory variables of interest. We also considered including an interaction between diarrhea and vomit in the model, however we excluded this interaction term from the final model due to issues with collinearity with the individual vomit and diarrhea variables. The final model is shown below:

$$\log R_{Eij} = \beta_0 + b_{0i} + \beta_1 Diarrhea_{ij} + \beta_2 Vomiting_{ij} + \beta_3 Resident_{ij} + e_{ij}$$

where log $R_{Eij}$ represents the estimated log $R_E$ of the $j^{th}$ case from the $i^{th}$ outbreak, $b_{0i}$ represents

the random slope for the $i^{th}$ outbreak, and $e_{ij}$ represents residual heterogeneity of the $j^{th}$ case from the $i^{th}$ outbreak not explained by the model. The residual heterogeneity, $e_{ij}$, and random slope, $b_{0j}$, are assumed to be independent and identically distributed (iid) with mean zero and their respective variances. Cases from the same outbreak were assigned the same random effect, whereas cases from different outbreaks were assumed to be independent. Final coefficient estimates and 95% confidence intervals were exponentiated to show the relationships between average $R_{Ei}$ (rather than log $R_{Ei}$) and the variables in the model. All statistical analyses were performed using the *EpiEstim* [22] and *metafor* [23] packages in R software version 3.4.2.

## Exclusion criteria

The original dataset consisted of 209 lab-confirmed and probable cases from six separate outbreaks. One case was excluded from the estimations of $R_{Ei}$ and all further analyses because he/she was missing an illness onset date. After the estimations of $R_{Ei}$, four additional cases were excluded from the regression analyses because they were missing information on diarrhea, vomiting, and/or resident vs. staff. Lastly, 9 more cases (4.3% of all cases with onset date information) had symptom onset dates on the last day the outbreak and thus did not produce any reported secondary cases. Therefore, they had estimated $R_{Ei}$'s of zero. Because log $R_{Ei}$ could not be taken for these cases, they were excluded from all regression analyses. Sensitivity analyses were performed by adding 0.01 to these $R_{Ei}$ estimates to examine the influence of these cases on model estimates.

## Results

Across the six outbreaks, the median number of cases was 36.5 (IQR: 28.3, 44.8) and the median outbreak length was 12 days (IQR: 12.0, 12.8) (Table 1). All cases involved in the outbreaks were either nursing home residents or staff. We have no data indicating that visitors were involved in the outbreaks. The majority of cases were over 80 years of age (62%), female (74%), nursing home residents (76%), and had diarrhea (with or without vomiting) (79%), vomiting (with or without diarrhea) (75%), or both diarrhea and vomiting (54%). Of the 9 cases excluded from regression analyses for having $R_{Ei} = 0$, 55% were residents, 55% reported vomiting, and 55% reported diarrhea. All six outbreaks were caused by norovirus genogroup II, two of which were confirmed as GII.4 Sydney and four of which were not genotyped.

**Table 1. Characteristics of analyzed nursing home norovirus outbreaks; South Carolina, 2014–2016.**

| Out-break No. | Total Cases, No. | Cases Excluded [a], No. (%) | Lab-confirmed Cases, No. | Outbreak Length (in days)[b] | Age (in y) Mean (SD) | Female, No. (%)[c] | Resident, No. (%)[c] | Diarrhea, No. (%)[c] | Vomit, No. (%)[c] |
|---|---|---|---|---|---|---|---|---|---|
| 1 | 27 | 1 (4) | 3 | 12 | 79 (17) | NA[d] | 23 (85) | 27 (100) | 19 (70) |
| 2 | 11 | 1 (9) | 4 | 10 | 84 (10) | 8 (73) | 11 (100) | 6 (55) | 10 (91) |
| 3 | 46 | 5 (11) | 4 | 13 | 83 (9) | 31 (67) | 38 (83) | 34 (76) | 28 (61) |
| 4 | 52 | 4 (8) | 4 | 18 | 88 (6) | 29 (74) | 44 (85) | 47 (92) | 49 (96) |
| 5 | 32 | 2 (6) | 4 | 12 | 84 (16) | 24 (75) | 20 (67) | 28 (88) | 22 (69) |
| 6 | 41 | 1 (2) | 4 | 12 | 81 (14) | 22 (85) | 22 (54) | 22 (54) | 27 (66) |
| Total[e] | 208 | 14 (7) | 23 | NA | 83 (12) | 114 (74) | 158 (76) | 164 (79) | 155 (75) |

[a]Cases excluded from the regression.

[b]Outbreak length is the difference in days between first illness and last illness onset dates (including the first illness onset date).

[c]Percentages were calculated excluding cases with missing information.

[d]Information on case sex was not collected for outbreak 1.

[e]One case with an unknown onset date was excluded from analyses.

Of the six outbreaks, four occurred during the 2014–2015 norovirus season (total cases: 141) and two occurred during the 2015–2016 norovirus season (total cases: 68). Due to the limited number of outbreaks, we were unable to make formal comparisons between seasons. However, a greater proportion of cases reported diarrhea, vomiting, and were nursing home residents in the 2014–2015 season (78%, 81%, and 83%, respectively) compared to the 2015–2016 season (68%, 66%, and 72%, respectively) and average $R_{Ei}$ estimates were similar (2014–2015: 0.95; 2015–2016: 0.97). Outbreaks began with one to three index case(s) (nine index cases in total), defined as cases with onset of symptoms on day one of an outbreak, that had large estimated $R_{Ei}$'s (range: 1.48 to 8.70) relative to other cases in the outbreak. After the index case(s), each outbreak either continuously declined to a $R_{Ei}$ below 1 or increased again before declining to a $R_{Ei}$ below 1 (Fig 1). Of these index cases, at least one from each outbreak reported vomiting (Fig 2). While most index cases also reported diarrhea, outbreak 6 began with a case that reported vomiting only.

When examining $R_{0i}$ values (calculated from $R_{Ei}$ estimates), we found that outbreaks had considerably higher basic reproduction numbers based on the index case(s) ($R_{0,1}$ = 6.8, 1.5, 8.4, 7.3, 4.6, and 8.7 for outbreaks 1–6, respectively) compared to the median basic reproduction number calculated from cases on days 2 to 4 (median $R_{0,2-4}$ = 1.7; IQR: 1.6, 2.0).

Cases with vomiting (with or without diarrhea) had a greater median $R_{Ei}$ (0.54; IQR: 0.21, 1.01) than those without vomiting (0.36; IQR: 0.20, 1.47), however the interquartile ranges were largely overlapping. Cases with diarrhea (with or without vomiting) had a similar median $R_{Ei}$ (0.45; IQR: 0.20, 1.01) to those without diarrhea (0.47; IQR: 0.27, 0.82). Cases with both vomiting and diarrhea had a greater median $R_{Ei}$ (0.78; IQR: 0.21, 1.03) than those with

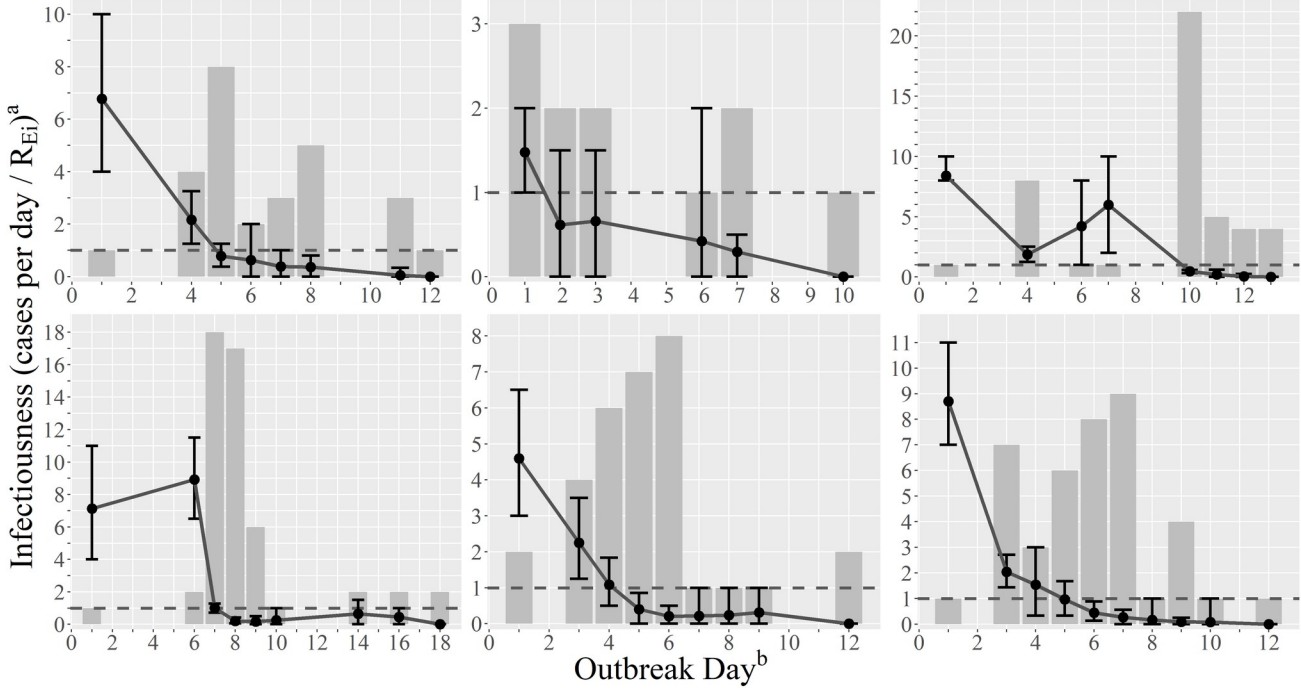

**Fig 1. Case counts and individual reproduction numbers, $R_{Ei}$, by day in nursing home norovirus outbreaks.** From left to right, outbreaks 1–3 and 4–6 are presented on top and bottom, respectively. Case counts are represented by the gray bars and $R_{Ei}$ estimates are represented by the point estimates with corresponding 95% confidence intervals. The horizontal dashed line signifies a $R_{Ei}$ of 1, below which each infectious individual, on average, infects less than one individual and the outbreak cannot be maintained. [a]Infectiousness describes the number of cases per day (for the gray bars) and $R_{Ei}$ (for the point estimates); note the change in scale for different outbreaks. [b]Outbreak day represents the day into the outbreak, with day 1 corresponding to the first illness onset date.

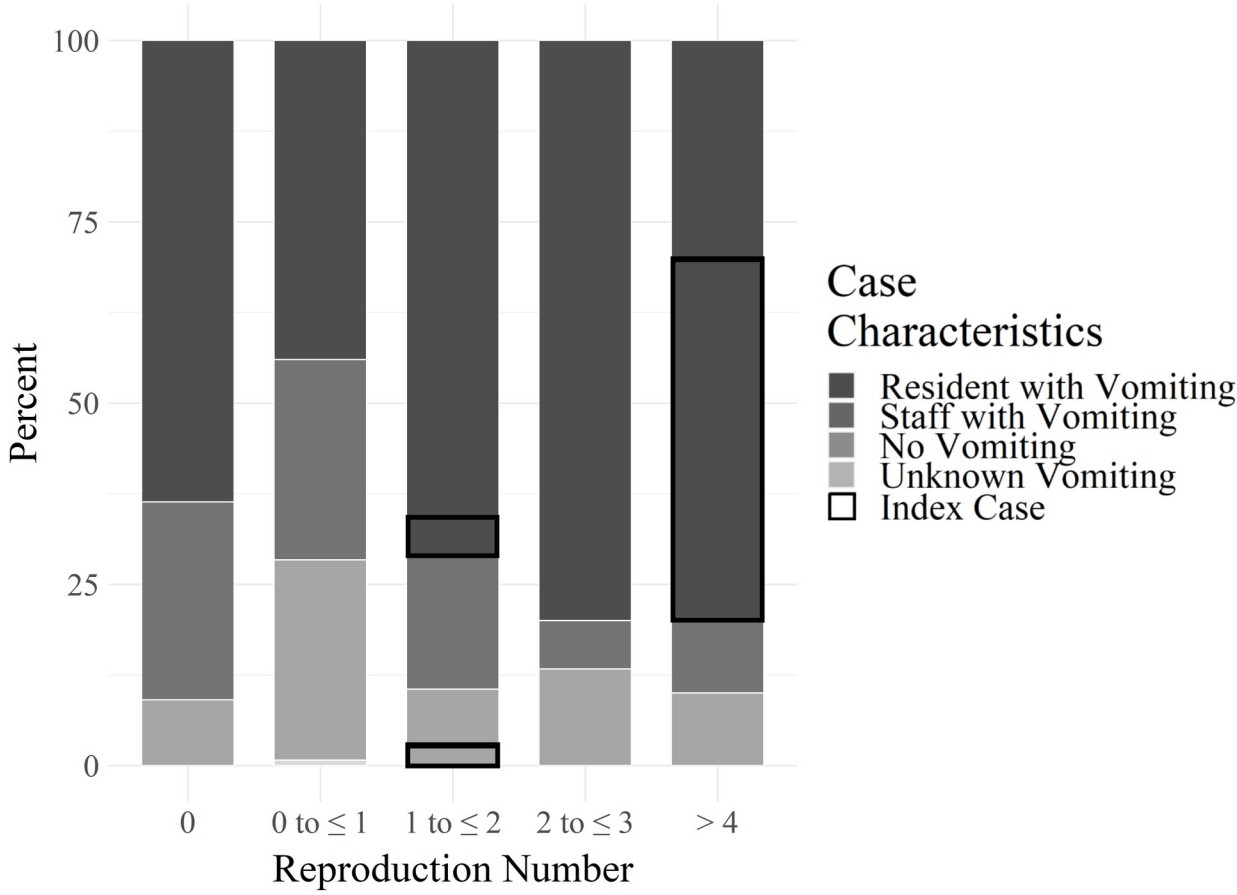

**Fig 2. Percentage of cases by individual reproduction number[a], $R_{Ei}$, for the following categories: residents who vomited, staff who vomited, residents/staff who did not vomit, and residents/staff with unknown vomiting; index cases are outlined in black.** [a]Individual reproduction number describes the number of secondary cases generated by an infectious case.

diarrhea alone (0.36; IQR: 0.20, 0.47) or vomiting alone (0.47; IQR: 0.27, 0.97), and residents had a slightly greater median $R_{Ei}$ (0.47; IQR: 0.21, 1.01) than staff (0.40; IQR: 0.21, 0.97). Inter-quartile ranges were again overlapping for all comparisons. Because all outbreaks ended, the overall median $R_{Ei}$ for all cases was less than 1 (0.47; IQR: 0.21, 1.01). Similarly, the median $R_{Ei}$ values for each outbreak were also less than 1, ranging from 0.40 to 0.63.

A total of 63 cases (30% of all cases) had an estimated $R_{Ei}$ greater than 1, of which 89% reported vomiting, 83% reported diarrhea, and 86% were residents. Among the remaining 145 cases (70% of all cases) with an estimated $R_{Ei}$ of less than 1, 68% reported vomiting, 77% reported diarrhea, and 71% were residents. All index cases had $R_{Ei}$'s greater than 1 (median: 4.60; IQR: 1.48, 7.13).

In the final multivariable model, cases who vomited infected 2.12 (95% CI: 1.68, 2.68) times the number of individuals as non-vomiters, cases with diarrhea infected 1.39 (95% CI: 1.03, 1.87) times the number of individuals as cases without diarrhea, and resident-cases infected 1.53 (95% CI: 1.15, 2.02) times the number of individuals as staff-cases (Fig 3). In sensitivity analyses where cases with $R_{Ei} = 0$ were included in the regression analysis, stronger associations between infectiousness and vomiting, diarrhea, and resident/staff status were observed: 2.56 (95% CI: 2.40, 2.73), 1.81 (95% CI: 1.69, 1.93), and 3.27 (95% CI: 3.05, 3.51), respectively.

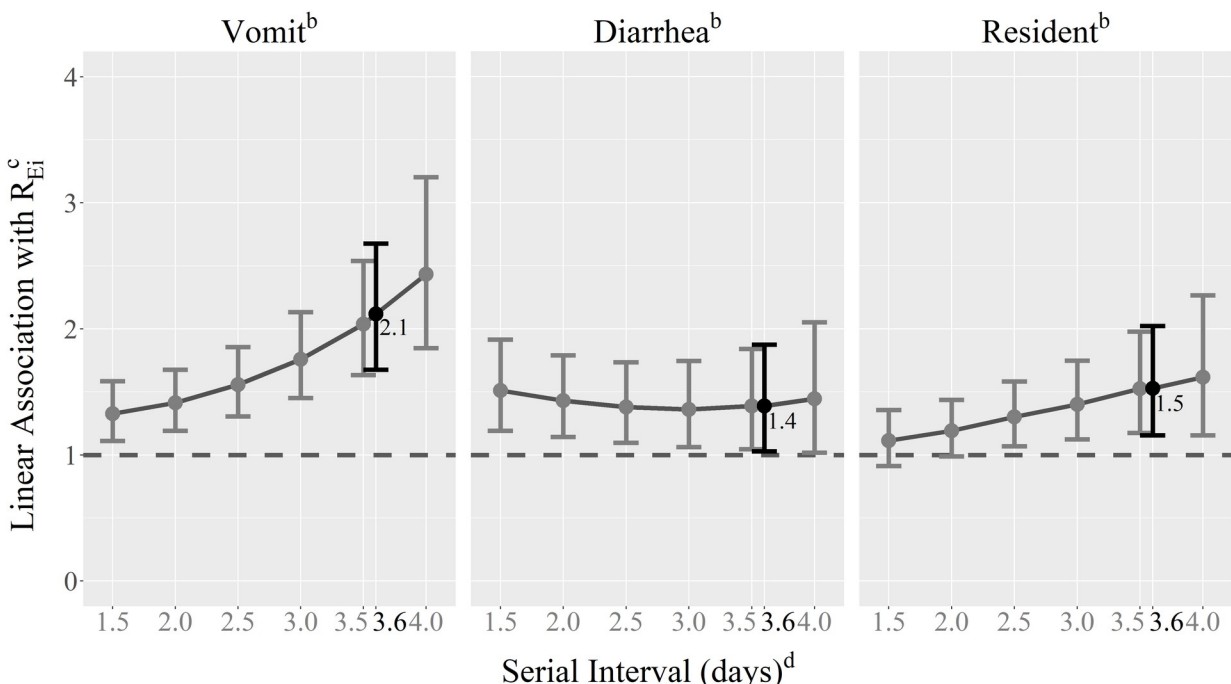

**Fig 3. Associations between individual reproduction numbers, $R_{Ei}$, and symptoms/characteristics of norovirus cases by serial interval length[a].** [a]The serial interval length used in the final regression analysis is shown in black. [b]Associations were estimated using a mixed linear regression model with a log-transformed outcome variable ($R_{Ei}$), inverse-variance weighting, a random slope for outbreak number, and the following dichotomous predictor variables: vomiting (vs. no vomiting), diarrhea (vs. no diarrhea), and resident (vs. staff). [c]Estimates from the model were exponentiated. [d]Estimates using a serial interval of 1.0 with a standard deviation of 2.0 (or 1.0) were unstable and therefore not reported.

Adding a dichotomous variable (index vs. non-index case) to the model indicated that index cases infected 3.96 (95% CI: 2.44, 6.44) times the number of individuals as non-index cases, holding resident vs. staff, diarrhea, and vomiting constant. Furthermore, we examined the associations between outbreak day, counting the first illness onset date as day one, and case characteristics and found cases who vomited occurred 2.7 (95% CI: 1.9, 3.5) days earlier in the outbreak than cases who did not vomit, cases with diarrhea occurred 1.8 (95% CI: 0.9, 2.6) days earlier in the outbreak than cases without diarrhea, and resident-cases occurred 1.6 (95% CI: 0.8, 2.3) days earlier in the outbreak compared to staff-cases.

In sensitivity analyses to examine the effect of using different norovirus serial intervals (serial intervals shorter and longer than 3.6 days) when calculating $R_{Ei}$, we found that associations between vomiting and $R_{Ei}$ and, to a lesser degree, resident and $R_{Ei}$ increased as the serial interval increased. The association between diarrhea and $R_{Ei}$ did not appear to change when the assumption about serial interval length was changed (Fig 3). Furthermore, in sensitivity analyses including sex and/or age in the model, we found limited evidence for confounding of the association between vomiting and case infectiousness. When sex and age were included in the model, the association was slightly attenuated: cases who vomited infected 1.90 (95% CI: 1.36, 2.66) times the number of individuals as non-vomiters. However, conclusions about the relative importance of vomiting remained the same. Associations between case infectiousness and diarrhea and resident largely disappeared: cases with diarrhea infected 1.09 (95% CI: 0.77, 1.55) times the number of individuals as cases without diarrhea and resident-cases infected 0.67 (95% CI: 0.25, 1.76) times the number of individuals as staff-cases, suggesting possible confounding by sex and/or age. However, in univariate analyses between the outcome variable,

exposure variables, and potential confounders (sex and age), only the association between resident and age was statistically significant.

## Discussion

We inferred who infected whom from outbreak line lists and investigated risk factors for transmission of norovirus in nursing home outbreaks, leading to several important findings. First, vomiting and, to a lesser degree, diarrhea play a critical role in norovirus transmission in these settings. Second, outbreaks tend to start with one or more cases who infect substantially more individuals than later cases in the outbreak. Third, residents, rather than staff, are the primary drivers of transmission. Our findings are based on data from multiple outbreaks affecting a considerable number of cases. We assumed the following: transmission of infection occurred only among reported cases, asymptomatic cases did not play a role in transmission, all reported cases were part of the same outbreak, and the serial interval for norovirus in this setting is gamma distributed with a mean of 3.6 days. Findings were generally robust to assumptions about the serial interval and inclusion/exclusion criteria for cases with missing data. While conclusions about the importance of vomiting in transmission remained the same when sex and age were included in the model, potential confounding of associations between case infectiousness and resident and diarrhea by sex and age should be further explored.

While previous studies have found that exposure to vomit is associated with an increased risk of norovirus infection in nursing home residents and staff [14], and that proximity to a vomiting event is correlated with higher attack rates [24,25], this is the first study to find that individuals, particularly residents, who vomit are more infectious and tend to drive norovirus transmission in U.S. nursing home outbreaks. Human challenge studies have found that vomiting, compared to diarrhea, is more likely to result in environmental contamination potentially leading to transmission through fomites and airborne droplets [26]. In household norovirus outbreaks, however, primary cases with diarrhea, but not vomiting, have been associated with higher secondary attack rates [27]. This suggests that the relative importance of specific symptoms in norovirus transmission may be dependent on the outbreak setting.

There is little systematic information available on norovirus introduction into nursing homes [14]. Outbreak reports have shown that nursing home outbreaks often start with single index cases [14], however the relative infectiousness of index cases (compared to non-index cases) has not been examined in these settings. We found that outbreaks tend to start with one or more cases who infect substantially more individuals compared to later cases. There are multiple possible explanations for this greater infectiousness of index cases. First, as an outbreak progresses and more individuals become ill and later immune, there is a natural decrease in the reproduction number. However, we found that index cases generally had substantially greater $R_{Ei}$'s compared to cases with onset dates only a few days after outbreak initiation, before a sufficient number of susceptibles could accumulate to explain this pattern. We also found that $R_{0,1}$ (the basic reproduction number for index cases) tended to be substantially larger than $R_{0,2-4}$ (the basic reproduction numbers for cases on days 2–4), even under the extreme assumptions that all individuals were initially susceptible and that the total population consisted only of reported cases in the outbreak. If the observed declines in $R_{Ei}$ had been due to a natural decrease in susceptibles alone, we would expect the calculated $R_{0i}$ values to remain relatively constant over time. Because this was the most extreme assumption, the differences between $R_{0,1}$ and $R_{0,2-4}$ estimates became even more pronounced when the population was assumed to have some level of immunity (e.g., 50% susceptibility). Therefore, these results suggest that index cases are more infectious than subsequent cases for reasons other than the natural decreases in susceptibles alone. Second, index cases may have been more infectious than

non-index cases due to intrinsic case characteristics (e.g., vomiting). Under this hypothesis, the median $R_{Ei}$ may be ~1.0, meaning that most cases in the outbreak are only moderately infectious, but a highly infectious case is required to initiate an outbreak. This hypothesis is supported by a recent paper that found that vomiting in norovirus index cases was associated with an increased risk of nosocomial outbreaks [28]. Third, rapid implementation of effective outbreak control measures could curtail transmission. Lacking data on the timing and type of control measures, we could not explicitly account for this in our calculations. Results may be due to any one of these explanations, or some combination thereof.

U.S. nursing home residents have an increased risk of norovirus gastroenteritis [8,14], but evidence for their relative infectiousness compared to staff was lacking. While staff clearly can transmit norovirus [12,14], studies of nosocomial outbreaks in the Netherlands have shown that symptomatic patients have the largest contribution to virus transmission in those settings [29]. The role of residents (vs. staff) in norovirus transmission in U.S. nursing homes may depend on the average level of mobility and dependency of residents. If nursing home residents are generally mobile, self-sufficient, and able to gather in communal rooms, they may be more likely than staff to contribute to norovirus transmission. Conversely, more self-sufficient residents may be more likely to identify symptoms of norovirus and self-quarantine, thus transmitting less. We did not have information on residents' mobility or dependence on nursing care for this study, so were unable to include these variables in our analyses.

We note a number of limitations of our study. First, all analyzed outbreaks took place in South Carolina, so results may not be generalizable to norovirus outbreaks in nursing homes in other U.S. states or elsewhere. Nursing home staffing levels vary widely across states [30], as do infection control training resources and healthcare-associated infection reporting [31]. Additionally, because all outbreaks were caused by genogroup II, and the two outbreaks genotyped were caused by GII.4 Sydney, results may not be generalizable to non-genogroup II or non-GII.4 outbreaks.

Second, symptomatic cases may go unreported, particularly in the early stages of an outbreak, which could lead to an overestimate of the infectiousness of index cases. Relatedly, index cases may have been misclassified if true index cases were missed. For norovirus, it's also possible that asymptomatic cases could contribute to transmission. However, studies have found that asymptomatic individuals in nosocomial outbreaks contribute significantly less to transmission than symptomatic individuals, leading us to believe that bias from missing asymptomatic cases is likely minimal [29]. Additionally, some reported cases could be sporadic or caused by a different etiologic agent. Furthermore, only the date of symptom onset, not time, was considered when calculating $R_{Ei}$'s. Because norovirus has a relatively short incubation period, it is possible, although unlikely, for primary and secondary cases to have the same symptom onset date. The method we used to calculate $R_{Ei}$ assumes that such cases cannot infect each other. Third, we excluded 9 cases with $R_{Ei} = 0$ from the regression analyses, however including them in the main regression analysis only strengthened the associations between all three predictor variables and infectiousness. The association between vomiting and increased infectiousness remained the strongest. Lastly, we used a serial interval distribution estimated from household transmission associated with norovirus outbreaks in child daycare centers in Sweden and assumed a similar serial interval in our U.S. nursing homes. Unlike transmission in the households, where it was clear that the child daycare center attendee/staff infected others in the home, identifying transmission pairs in nursing home outbreaks is difficult, precluding direct estimation of serial intervals in these settings. The true serial interval may be longer or shorter in nursing homes. Regardless, we found that our main finding of the importance of vomiting in transmission was robust when using different values of the serial interval.

Because there is currently no publicly available norovirus vaccine, sound prevention and control measures are key to controlling norovirus outbreaks, but the present body of published literature does not provide an evidence-base for the value of specific measures [10]. These study results lend support for measures that focus on cases who vomit, particularly if those cases are residents (vs. staff). Results indicate that rapid response to a vomiting event may be effective in reducing the size and duration of norovirus outbreaks in nursing home settings, and support measures that reduce exposure to vomit, such as thorough cleaning and disinfection with a chlorine-based disinfectant, isolation of the case, and implementing antiemetic treatment after the first vomiting episode [26]. While cleaning with a chlorine-based disinfectant and isolation of cases are currently recommended control strategies, there are generally little data about the implementation and timing of these control measures in norovirus outbreaks [10]. Furthermore, the use of antiemetic drugs for the prevention and control of norovirus outbreaks is not widely recommended [32,33]. The important role of vomiting in transmission shown in our results suggests that the use of antiemetic drugs in nursing home outbreaks should be further considered. Information on type and timing of control measures was not available for this study. Future studies should collect such data and evaluate the effects of specific control measures using similar analytical methods to the approach used here. Future studies should also further examine the association between case infectiousness and the interaction between vomiting and diarrhea.

## Conclusions

Vomiting, particularly by residents, drives norovirus transmission in U.S. nursing home outbreaks. This has implications for prevention and control measure recommendations for outbreaks in these settings.

## Supporting information

**S1 File. Outbreak Information.** This file contains information on all cases involved in the six norovirus outbreaks presented in the paper, including individual reproduction number estimates for each case.
(CSV)

**S1 RMarkdown. Estimating Individual Reproduction Numbers.** This file contains instuctions and R code for estimating individual reproduction numbers.
(HTML)

**S2 RMarkdown. Serial Interval Sensitivity Analysis.** This file contains instuctions and R code for estimating individual reproduction numbers using different serial interval lengths.
(HTML)

**S3 RMarkdown. Regression Analyses.** This file contains instuctions and R code for running weighted mixed regression analyses.
(HTML)

**S1 Poster. NoroCORE Poster Presentation.** This poster shows preliminary results from the paper and was presented in March 2018 at the NoroCORE Final Showcase Meeting in Atlanta, GA, USA.
(PDF)

## Acknowledgments

The findings and conclusions in this report are those of the authors and do not necessarily represent the official position of the Centers for Disease Control and Prevention.

## Author Contributions

**Conceptualization:** Carly Adams, Benjamin A. Lopman.

**Data curation:** Carly Adams, David Young.

**Formal analysis:** Carly Adams.

**Funding acquisition:** Benjamin A. Lopman.

**Investigation:** Carly Adams.

**Methodology:** Carly Adams, Paul A. Gastañaduy, Prabasaj Paul, Zach Marsh, Aron J. Hall, Benjamin A. Lopman.

**Supervision:** Benjamin A. Lopman.

**Writing – original draft:** Carly Adams.

**Writing – review & editing:** Carly Adams, David Young, Paul A. Gastañaduy, Prabasaj Paul, Zach Marsh, Aron J. Hall, Benjamin A. Lopman.

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
