## [Decision Letter · Decision Letter 0]

10 Oct 2019

Dear Dr Adams,

Thank you very much for submitting your manuscript 'Quantifying the roles of vomiting, diarrhea, and residents vs. staff in norovirus transmission in U.S. nursing home outbreaks' for review by PLOS Computational Biology. Your manuscript has been fully evaluated by the PLOS Computational Biology editorial team and in this case also by independent peer reviewers. The reviewers appreciated the attention to an important problem, but raised some substantial concerns about the manuscript as it currently stands. While your manuscript cannot be accepted in its present form, we are willing to consider a revised version in which the issues raised by the reviewers have been adequately addressed. We cannot, of course, promise publication at that time.

Sincerely,

Roger Dimitri Kouyos

Associate Editor

PLOS Computational Biology

Rob De Boer

Deputy Editor

PLOS Computational Biology

[LINK]

Reviewer's Responses to Questions

**Comments to the Authors:**

Reviewer #1: Please see attachment

Reviewer #2: Overview

The authors analyse data from six outbreaks of norovirus in long-term care facilities in South Carolina, USA. They estimate individual reproduction numbers and link these to factors such as displayed symptoms and whether the individual was a resident or staff. They find strong indications that vomiting increases individual reproduction numbers and that residents generally infect more individuals per generation than staff. The authors suggest that vomiting may drive transmission of norovirus in nursing homes in the USA.

This paper is well written and clear, the supplementary information is particularly well-presented. The analyses rely on established R packages to estimate reproduction numbers for individuals in each outbreak. The results are first to quantify a link which has been established in other studies, that vomiting increases risk of transmission. A few points of clarification concerning the population immunity are requested below.

Detailed comments

Line 111: As the outbreaks come from two seasons- is there any evidence for different transmission dynamics between the two?

Line 131: Estimates of Heijne et al. go up to R0 ~14.

Line 150: It is worth noting that index cases, due to their limited nature (ie. there will be only 1-3 per outbreak) will have the most uncertain Rei. In this way, including further outbreaks in this analysis may produce a better estimate of how infectious index cases are compared to other cases (not a suggestion for this paper).

Line 154: Stating that the entire population is susceptible is a strong assumption, as mentioned in the discussion. It is difficult to establish pre-existing immunity; I would suggest a sensitivity analysis exploring the possibility that eg. 90% of the nursing home population is susceptible at the start.

Line 187: The outbreaks are assumed to be independent but they all occurred in South Carolina within two years. Could you comment on the likelihood that the outbreaks may be related?

Line 190: Please add a note to explain the Kruskal-Wallis test.

Line 216: Were the excluded cases distributed evenly between different outbreaks?

Line 222: Many of the cases were not lab confirmed for norovirus, is it safe to assume these are from the same outbreak?

Line 228: Index cases are highly uncertain, can you comment more on the reporting of the index case ie. what is the chance that earlier cases were missed?

generally: Asymptomatic infection: how would this affect the estimates?

Fig 1: The bottom left outbreak has a break between the index and secondary cases, how likely is that intermediate cases were missed?

Fig 2: Generally the paper and supplementary markdown documents are presented well. As such, I think this figure could be improved. Perhaps with index cases as dots rather than boxes. Additionally, with the grey scale bars, it may be better to have a white background- something like theme_bw().

Line 249: The conclusion that index cases are more infectious could be explained by other factors. These cases are most uncertain due to reporting and will be subject to uncertainty due to the assumptions concerning the population immunity and changing in control efforts.

Line 286: Any comment on why the influence of resident status should be affected by serial interval?

Line 387: Are there current guidelines for control of vomiting to compare these suggestions to?

Reference 22: Appears incorrectly formatted

Reviewer #3: This study estimates the impact of a few variables (vomiting, diarrhea and resident vs staff status) on the infectiousness of norovirus cases. The study is based on quite limited data from a number of norovirus outbreaks, but is relevant nonetheless. The methods are valid and may be of interest to other researchers.

Major comments:

None

Minor comments:

Intro: the distinction between LTCFs and nursing homes may need to be clarified.

Some limitations of the study merit mentioning:

• Absence of genotype information from some outbreaks (and thus the generalizability to all norovirus outbreaks)

• Absence of data on age and gender due to a large amount of missing data. This is problematic as some of the observed associations may be linked to age (and less probably gender). Can the authors not check this using only the cases with available data?

**Have all data underlying the figures and results presented in the manuscript been provided?**

Reviewer #1: Yes

Reviewer #2: Yes

Reviewer #3: Yes

PLOS authors have the option to publish the peer review history of their article (what does this mean?). If published, this will include your full peer review and any attached files.

Reviewer #1: No

Reviewer #2: No

Reviewer #3: Yes: Thomas Verstraeten, MD

---

## [Decision Letter · Decision Letter 1]

6 Jan 2020

Dear Dr Adams,

Thank you very much for submitting your manuscript, 'Quantifying the roles of vomiting, diarrhea, and residents vs. staff in norovirus transmission in U.S. nursing home outbreaks', to PLOS Computational Biology. As with all papers submitted to the journal, yours was fully evaluated by the PLOS Computational Biology editorial team, and in this case, by independent peer reviewers. The reviewers appreciated the attention to an important topic but identified some aspects of the manuscript that should be improved.

We would therefore like to ask you to modify the manuscript according to the review recommendations before we can consider your manuscript for acceptance. Your revisions should address the specific points made by each reviewer and we encourage you to respond to particular issues Please note while forming your response, if your article is accepted, you may have the opportunity to make the peer review history publicly available. The record will include editor decision letters (with reviews) and your responses to reviewer comments. If eligible, we will contact you to opt in or out.raised.

- Supporting Information uploaded as separate files, titled 'Dataset', 'Figure', 'Table', 'Text', 'Protocol', 'Audio', or 'Video'.

We hope to receive your revised manuscript within the next 30 days. If you anticipate any delay in its return, we ask that you let us know the expected resubmission date by email at ploscompbiol@plos.org.

Sincerely,

Roger Dimitri Kouyos

Associate Editor

PLOS Computational Biology

Rob De Boer

Deputy Editor

PLOS Computational Biology

[LINK]

Reviewer's Responses to Questions

**Comments to the Authors:**

Reviewer #1: Please see attachment

Reviewer #2: I am happy to recommend accepting the article.

Reviewer #3: All comments have been addressed.

**Have all data underlying the figures and results presented in the manuscript been provided?**

Reviewer #1: Yes

Reviewer #2: Yes

Reviewer #3: None

PLOS authors have the option to publish the peer review history of their article (what does this mean?). If published, this will include your full peer review and any attached files.

Reviewer #1: Yes: Seth Blumberg

Reviewer #2: No

Reviewer #3: Yes: Thomas Verstraeten

---

## [Decision Letter · Decision Letter 2]

21 Feb 2020

Dear Ms. Adams,

We are pleased to inform you that your manuscript 'Quantifying the roles of vomiting, diarrhea, and residents vs. staff in norovirus transmission in U.S. nursing home outbreaks' has been provisionally accepted for publication in PLOS Computational Biology.

Before your manuscript can be formally accepted you will need to complete some formatting changes, which you will receive in a follow up email. A member of our team will be in touch within two working days with a set of requests.

Best regards,

Roger Dimitri Kouyos

Associate Editor

PLOS Computational Biology

Rob De Boer

Deputy Editor

PLOS Computational Biology

Reviewer's Responses to Questions

**Comments to the Authors:**

Reviewer #1: Thank you for your thoughtful reply and consideration of reviewer feedback. All my questions have been comprehensively addressed.

**Have all data underlying the figures and results presented in the manuscript been provided?**

Reviewer #1: Yes

PLOS authors have the option to publish the peer review history of their article (what does this mean?). If published, this will include your full peer review and any attached files.

Reviewer #1: Yes: Seth Blumberg

---

## [Editor Report · Acceptance letter]

18 Mar 2020

PCOMPBIOL-D-19-01180R2 

Quantifying the roles of vomiting, diarrhea, and residents vs. staff in norovirus transmission in U.S. nursing home outbreaks

Dear Dr Adams,

I am pleased to inform you that your manuscript has been formally accepted for publication in PLOS Computational Biology. Your manuscript is now with our production department and you will be notified of the publication date in due course.

With kind regards,

Sarah Hammond
